# A fast numerical method for oxygen supply in tissue with complex blood vessel network

**Yuankai Lu**, **Dan Hu***, **Wenjun Ying**

School of Mathematical Sciences, Institute of Natural Sciences, and MOE-LSC, Shanghai Jiao Tong University, Shanghai, China

* hudan80@sjtu.edu.cn

## Abstract

Angiogenesis plays an essential role in many pathological processes such as tumor growth, wound healing, and keloid development. Low oxygen level is the main driving stimulus for angiogenesis. In an animal tissue, the oxygen level is mainly determined by three effects—the oxygen delivery through blood flow in a refined vessel network, the oxygen diffusion from blood to tissue, and the oxygen consumption in cells. Evaluation of the oxygen field is usually the bottleneck in large scale modeling and simulation of angiogenesis and related physiological processes. In this work, a fast numerical method is developed for the simulation of oxygen supply in tissue with a large-scale complex vessel network. This method employs an implicit finite-difference scheme to compute the oxygen field. By virtue of an oxygen source distribution technique from vessel center lines to mesh points and a corresponding post-processing technique that eliminate the local numerical error induced by source distribution, square mesh with relatively large mesh sizes can be applied while sufficient numerical accuracy is maintained. The new method has computational complexity which is slightly higher than linear with respect to the number of mesh points and has a convergence order which is slightly lower than second order with respect to the mesh size. With this new method, accurate evaluation of the oxygen field in a fully vascularized tissue on the scale of centimeter becomes possible.

## 1 Introduction

Oxygen plays a key role in animal metabolism. Oxygen supply to tissue is mainly achieved by the circulation system of animals. In particular, the efficiency of oxygen delivery is mainly determined by the microcirculation structure. In order to improve the efficiency of their microcirculation structure, animals have developed different physiological processes to modify the geometry and topology of vessel networks, including blood vessel adaptation and angiogenesis [1–3]. In these physiological processes, the oxygen level is the key driving stimulus. For example, poor microcirculation structure results in local tissue hypoxia (low concentration of oxygen), which leads to the production of growth factors for angiogenesis, such as vascular endothelial growth factor (VEGF) [4, 5]. Angiogenesis plays a critical role in many pathological processes, such as wound healing [6], and keloid development [7], and tumor growth

**Data Availability Statement:** All relevant data are within the paper and its Supporting information files.

**Funding:** This work is supported by National Key R&D Program of China (2019YFA0709503), the

National Natural Science Foundation of China (Contract no. 11971312, 11771290, and 91630208). The funders had no role in study design, data collection and analysis, decision to publish, or preparation of the manuscript.

**Competing interests:** The authors have declared that no competing interests exist.

[8, 9]. Modeling studies of angiogenesis are important in understanding the progression of tumor [10] and its therapy [11–13]. Accurate evaluation of oxygen supply is also important in generating artificial vascular networks [14, 15] and non-invasively investigating the impact of diseases and therapeutic procedures on the vascular bed [16].

There have been a few techniques for measurement of oxygen level, such as two-photon phosphorescence lifetime microscopy that can be applied to measure oxygen level in *vivo* [17]. However, existing techniques can hardly offer a complete spatial-temporal picture of oxygen field on microscopic scales. Therefore, theoretic modeling [18–20] and numerical simulation [21–26] is widely utilized in evaluating oxygen level and studying angiogenesis.

The classical Krogh cylinder model roughly describes the oxygen transport from blood vessels to tissues [18, 27]. In this model, evenly spaced capillaries are assumed to be parallel and supply oxygen to a cylindrical tissue domain. Following Krogh's model, more detailed oxygen consumption mechanisms are taken into account [28]. Coupled models for oxygen delivery including a reasonable oxygen consumption mechanism, a relatively complex vessel network structure, and detailed blood flow in the vessel network are also proposed in later works [19, 29].

Experimental results have indicated disordered spatial distribution of blood vessels [30]. In retinal vascular network, the arterioles and venules reach out from the center to the periphery, forming a roughly radial skeleton, while capillaries form a vessel network that links the arterioles and venules [31]. It is also observed that the microcirculation in tumor tissue presents a chaotic geometry [32]. Therefore, in most real applications, it is important to incorporate the complex vessel network structures into the model and numerical algorithms to evaluate the oxygen field in tissue.

The chaotic vessel geometry brings great challenges in designing effective numerical algorithms. Although existing simulations based on finite difference method and rectangular grids can provide useful insight for tissue oxygen supply, they introduce unrealistic requirement for the geometry of blood vessel networks [2, 22]. A finite element method is also introduced for more general vessel network topology and geometry [28]. This method requires a particular local processing near blood vessels. Meanwhile, the mesh grid is generated according to the vessel geometry. As a result, the corresponding computational cost is huge for tissue with large-scale complex vessel network structure. In more realistic applications, Secomb and Hsu developed a numerical method based on Green's function, in which each vessel is regarded as a line source of oxygen [26, 32]. Their numerical method can be used to deal with complex vessel geometry. However, their method suffers from the high computational cost due to the all-to-all interaction between all elements. In order to handle large-scale simulations, Welter et al. introduced a fast numerical method based on finite element method [33]. With a uniform grid, they achieved the simulation in a fully vascularized tissue with a domain size of about 0.5 $cm^3$, which was used to study the pathological characteristics of a tumor and its surrounding tissues [34].

In this work, we develop a fast numerical method to evaluate the coupled system for oxygen delivery in tissue, with particular attention paid to the large-scale complex blood vessel network structures. In our model, the blood vessel is also regarded as a line source for oxygen field. A source distribution technique is used so that we can solve the oxygen field on a square mesh. This technique ensures that blood vessels can be freely embedded in the tissue without the need to fit the grid mesh. A post-processing technique is employed to remove the numerical error induced by source distribution, which allows us to use a relatively large mesh size for the square mesh, while sufficient numerical accuracy is maintained. Furthermore, an efficient iteration method is used to deal with the nonlinearity of the coupled system. The convergence order of our new method is slightly smaller than second order with respect to the mesh size

and the computational cost is slightly larger than linear with respect to the total number of mesh points. Due to the advantages of this new method, we can accurately evaluate the oxygen field of a three dimensional fully vascularized tissue on the scale of centimeter within $20h$.

## 2 Modeling

The model for oxygen delivery in blood vessels and tissues is well established in previous works [19, 29]. The model mainly includes a partial differential equation (PDE) for the oxygen field, a linear system for blood flow and blood pressure, and a system of ordinary differential equations (ODE) for the blood oxygen.

### 2.1 Oxygen diffusion in tissue

The oxygen-consuming tissue is a mixture of cells, extracellular matrix and extracellular fluid. Oxygen transport in tissue depends mostly on diffusion. The diffusion constant and solubility of oxygen may vary slightly in different tissue. Here we assume a uniform oxygen diffusivity $D$ and uniform solubility $\alpha$. Oxygen diffusion in tissue at steady state satisfies the reaction-diffusion equation

$$D\alpha \triangle P_{\mathrm{O}} = M(P_{\mathrm{O}}), \tag{1}$$

where $P_{\mathrm{O}}$ is the partial pressure of oxygen and $M(\cdot)$ is the oxygen-consuming rate in tissue. The oxygen consumption in cells consists of various biochemical processes, which can be described by the Michaelis-Menten equation in general

$$M(P_{\mathrm{O}}) = \frac{M_0 P_{\mathrm{O}}}{P_0 + P_{\mathrm{O}}}, \tag{2}$$

where $M_0$ represents the maximum consumption under infinite oxygen supply and $P_0$ represents the partial pressure of oxygen at half-maximal consumption.

### 2.2 Blood flow and blood pressure

The blood flow and blood pressure can be computed with Ohm's law and Kirchhoff's circuit law that form a system of linear equations. For a small blood vessel, which can be regarded as a cylinder, the blood flow in the vessel can be well approximated by the Poiseuille flow. The conductance of the vessel is

$$C_{ij} = \frac{\pi R_{ij}^4}{8\mu L_{ij}}, \tag{3}$$

where $i$ and $j$ are indices of the two end nodes of the vessel, $\mu$ is the viscosity of the blood, and $R_{ij}$ and $L_{ij}$ are the radius and length of the vessel, respectively. It is worth noting that, the effective viscosity $\mu$ can be dependent on the vessel radius and blood oxygen level [26, 32]. In this case, the system becomes a nonlinear system that is coupled to the whole system for oxygen delivery.

The blood pressure and blood flow rate from node $i$ to node $j$, $Q_{ij}$, satisfies the Ohm's law

$$Q_{ij} = -Q_{ji} = (P_i - P_j)C_{ij}.$$

The Kirchhoff's circuit law describes the conservation of mass

$$\sum_j Q_{ij} = s_i, \tag{4}$$

where $s_i$ gives the sources and sinks at the inlets and outlets of the vessel network. In most cases, fluid exchange between the blood and tissue may only be a small percent of the total flow (e.g., <0.5% as evaluated from Ref [35]). In this case, we can simply set $s_i = 0$ for all inner nodes including the junction points. When this fluid exchange becomes large, such as in the kidney or in particular pathological states, the Starling's law can be used by introducing continuous flow sinks $s_i$ along the vessels [35]. In real applications, the boundary conditions at the inlets and outlets can also be replaced by other conditions, such as fixed pressure condition.

### 2.3 Oxygen flux in vessels

Oxygen carried by blood includes two parts, the minor of which is directly dissolved in the plasma while the major of which is associated with hemoglobin. The relation between oxygen saturation $S_a$ and blood oxygen partial pressure $P_b$ satisfies the Oxygen-hemoglobin dissociation relation

$$S_a(P_b) = \frac{P_b^n}{P_b^n + P_{50}^n},$$
(5)

where $n = 2$ is used in this work and $P_{50}$ is the half-saturated oxygen pressure that may depend on the pH-value of blood. Correspondingly, the oxygen flux $f$ through a cross-section of the vessel includes two parts

$$f(P_b) = Q(\alpha_b P_b + H_D C_0 S_a(P_b)).$$
(6)

where $Q$ is the blood flow rate, $\alpha_b$ is the oxygen solubility in blood plasma, $H_D$ is the discharge hematocrit, and $C_0$ is the concentration of hemoglobin-bound oxygen in a fully saturated red blood cell (RBC).

### 2.4 Oxygen exchange on vessel walls

Let $s$ be the arc-length parameter along the centerline of a vessel and $\mathbf{x}(s)$ be the coordinate of the centerline. Assume the cross-section of the blood vessel is a circle with radius $R$, then the total oxygen flux $q(s)$ through the blood vessel wall per unit length is

$$
\begin{aligned}
q(s) &= -D\alpha \int_0^{2\pi} \frac{\partial P_O(r, \theta, s)}{\partial r}\Big|_{r=R} R d\theta \\
&= -D\alpha R \frac{\partial}{\partial r}\Big|_{r=R} \int_0^{2\pi} P_O(r, \theta, s, t) d\theta,
\end{aligned}
$$
(7)

where $r$ is the polar radius and $\theta$ is the polar angle on the cross-sectional plane (the pole is at $\mathbf{x}(s)$). In this work, we assume that the diffusion constant of oxygen in the blood vessel wall is the same as that in the tissue, in which the vessel wall can be regarded as part of the tissue. Therefore, the oxygen flux can be evaluated from the average $P_O$ gradient on the vessel wall. The oxygen flux can also be described by the Kedem-Katalchsky's law [36],

$$q(s) = 2\pi R L_p (P_b - P_{O,\text{wall}} - \sigma \Delta \Pi)$$
(8)

where $L_p$ is the hydraulic permeability of the vessel wall, $P_b(s)$ is the blood oxygen pressure, $P_{O,\text{wall}}(s)$ is the circumferential average oxygen partial pressure on the outer surface of the vessel wall, and $\sigma$ and $\Delta \Pi$ are the osmotic reflection coefficient and the osmotic pressure difference, respectively. In the Kedem-Katalchsky's law, precise experimental measurements of multiple parameters (e.g., the thickness of blood vessel wall and the vascular permeability) are used to evaluate the conductivity of oxygen through the blood vessel wall.

Ignoring the oxygen consumption in blood, we obtain the conservation of oxygen

$$\frac{df(P_{\text{b}})}{ds} = -q(s). \tag{9}$$

Given the oxygen flux $q(s)$, the above equation becomes a first-order ordinary differential equation (ODE). A boundary condition is required to solve the equation for each vessel. In real applications, we may have $P_{\text{b}}$ at the inlet of each vessel: (1) At the inlets of the vessel system, $P_{\text{b}}$ is given; (2) At the bifurcation points, $P_{\text{b}}$ at the inlet of the downstream vessels is inherited from the parent vessel; (3) At collecting junctions, $P_{\text{b}}$ at the inlet of the downstream vessel is obtained from its parent vessels as the mixed value by conservation of oxygen.

## 2.5 Model simplification

Since the oxygen partial pressure should be continuous in the whole domain, i.e., the tissue domain and the vessel domain, $P_{\text{b}}$ provides a Dirichlet boundary condition on vessel walls for $P_{\text{O}}$ in Eq (1). However, the disordered structure of blood vessel walls brings great difficulties in meshing and numerical simulations to solve the above coupled system. In previous studies [37], the oxygen flux is represented by oxygen sources on the centerlines of vessels. Under this simplification, the governing equation for oxygen supply is defined in the whole domain. In this case, the governing equation becomes

$$-D\alpha \triangle P_{\text{O}} \quad = \quad -M(P_{\text{O}}) + S(\mathbf{x}), \tag{10}$$

where $S(\mathbf{x}) = \int q(s)\delta(\mathbf{x} - \mathbf{x}(s))ds$ is the oxygen source supplied by the vessel.

Note that the vessel radius is relatively small ($2 \sim 3\mu m$) compared to the distance between capillaries ($\sim 100\mu m$). Therefore, this simplification can be a good approximation at least for far field. The oxygen partial pressure may be overestimated near the vessels, which can lead to a slight overestimate of the oxygen source $M(P_{\text{O}})$. Further improvement on such an approximation has been discussed in the work of Ref. [26, 38]. Instead of concentrated oxygen sources on the centerlines of blood vessels, distributed sources using smooth kernel functions are also used in a recent study [39]. This can be effectively used to avoid the weak singularities of the oxygen field.

Now we have obtained a coupled system for oxygen delivery, which includes a PDE (10) on the whole domain, a set of nonlinear ODEs (9) on the vessel centerlines, and a system of linear equations for blood pressures and blood flows in all vessels.

## 3 Numerical method

In order to develop a fast numerical method to solve the above coupled system, we are left with two main tasks: (1) find an efficient iteration method to deal with the nonlinearity of the coupled system and (2) develop a fast numerical solver for the PDE (10) with complex sources on blood vessel centerlines.

### 3.1 Nonlinear iteration

Mainly due to the nonlinearity of the ODEs (9) and the unusual form of coupling, general iteration methods, such as the full Newton-like iterative methods, are both hard in code implementation and lack of convergence guarantee for the coupled system. Here we propose an iterative method by decoupling the system and alternatively updating $P_{\text{O}}$ and $P_{\text{b}}$. In particular, we introduce pseudo time step in order to reach the steady state solution. The rest of the

section then explains how each (pseudo) time step is solved, namely with a semi-implicit Euler method.

Given $P_O^n$ and $P_b^n$ at step $n$, by assuming that the blood oxygen partial pressure $P_O = P_b$ on vessel walls, Eq (7) allows us to evaluate the flux $q^n(s)$ numerically. In the 3-D case, we expand the Laplacian operator in series near the vessel first, and then use particular fitting to evaluate the derivative in Eq (7). In the 2-D case, linear fitting on both sides of the vessel can be directly utilized to evaluate the flux $q^n(s)$. Detailed discussion on the fitting method is included in Appendix A. Similar treatment can also be used to evaluate the oxygen flux utilizing the Kedem-Katalchsky's law. In general, we denote the evaluation of $q^n$ as

$$q^{n*}(s) \quad = \quad L(P_b^n, P_O^n), \tag{11}$$

which is linearly dependent on $P_O^n$ and $P_b^n$. According to our numerical tests, it is not efficient for convergence to update the flux directly using $q^{n*}$. Instead, a weighted sum

$$q^{n+1}(s) \quad = \quad \lambda q^n(s) + (1-\lambda)q^{n*}(s) \tag{12}$$

is more efficient, where the weight $\lambda = \frac{1}{2}$ is used in our simulation. Once the flux is obtained, Eq (9) is used to update $P_b$ in all vessels

$$\frac{df(P_b^{n+1})}{ds} = -q^{n+1}(s). \tag{13}$$

The fourth-order Runge-Kutta method is used to solve the ODE (13) to obtain $P_b^{n+1}$ in each vessel segment. In practice, $q^{n*}$ can also be updated in a Gauss-Seidel fashion. Namely, the values of $P_b^{n+1}$ on the updated nodes can be used in Eq (11) to evaluate the flux.

The flux $q^{n+1}$ is also used to calculate the oxygen source $S^{n+1}$, which is required in computing $P_O^{n+1}$. It is possible to update $P_O$ by fully solving the PDE (10) with given oxygen source $S^{n+1}$. However, since we still need the nonlinear iteration, it is neither necessary nor efficient to find the accurate solution at each iteration step. Instead, we use the following scheme to update $P_O$

$$\frac{1}{\Delta t}(P_O^{n+1} - P_O^n) = D\alpha\triangle_h P_O^{n+1} - M(P_O^{n+1}) + S^{n+1}, \tag{14}$$

where $\triangle_h$ is the numerical Laplacian operator and $\Delta t$ is the temporal step size. This implicit numerical scheme solves the time-dependent reaction-diffusion equation for one time step. When the iteration converges, the solution satisfies the steady state PDE (10).

Note that an increase of the oxygen partial pressure $P_O^n$ in tissue can lead to a decrease of the oxygen flux $q^{n+1}$ (thus the oxygen source $S^{n+1}$). This implies at least one negative eigenvalue of the coupled system in the sense of linearization. As a result, it may even be not stable to update $P_O$ by fully solving the steady state PDE (10) for each iteration step (namely, $\Delta t = \infty$). The instability has been observed in our numerical test. From this point of view, a suitable time step size $\Delta t$ should be selected so that it is both good for iteration stability ($\Delta t$ is not too big) and good for fast convergence ($\Delta t$ is sufficiently big).

## 3.2 PDE solver

Based on the iteration framework above, our remaining task is to numerically solve the PDE (10) efficiently. For the simplified coupled system, we do not consider the detailed geometry of vessel walls. Thus, a square mesh can be easily used in our simulation. We simply use the central difference as the numerical Laplacian $\triangle_h$, where $h$ is the mesh size. This brings great convenience in developing a fast solver. The system in this paper contains no fluid convection and

maintains a uniform diffusion coefficient. Under this situation, the central differential scheme is equivalent to the finite volume method, which ensures the local mass conservation. When convection becomes significant, we can use the mass-conservative finite volume method instead.

There are two problems left for us to solve: First, the oxygen sources are located on vessel centerlines, we need to distribute the sources onto the square mesh points; Second, in order to conduct a large scale simulation, we need to use a relatively large spatial mesh size while maintaining sufficient numerical accuracy.

In order to distribute the oxygen sources onto the square mesh points, first we discretize the sources to be point sources $S_k^n(\mathbf{x}) = q^n(\mathbf{x}_k)h_k\delta(\mathbf{x} - \mathbf{x}_k)$ on the center lines, where $k$ is the index of the point source at $\mathbf{x}_k = (x_{k,1}, x_{k,2}, \ldots, x_{k,d})$ ($d$ is the space dimension) and $h_k$ is the step size on the centerline; Then, we use Peskin's numerical $\delta$-function $\bar{\delta}(\cdot)$ (see Appendix B) to distribute all point sources onto their neighboring mesh points [40]. Namely, we have

$$\bar{S}_{\mathbf{i}}^n = \sum_k q^n(\mathbf{x}_k)h_k\bar{\delta}(\mathbf{x} - \mathbf{x}_k), \qquad (15)$$

where $\bar{\delta}(\cdot)$ is the numerical $\delta$-function and $\mathbf{i} = (i_1, i_2, \ldots, i_d)$ is the index of the mesh point.

Due to the nonlinearity in the consumption function $M(\cdot)$, we use the standard multigrid algorithm combined with the Newton's method to solve Eq (14). According to our numerical tests, only a few steps of Newton-iteration are sufficient to make the numerical error small enough.

Notably, the above redistribution of oxygen-sources can lead to numerical error in the solution. As a result of using Peskin's numerical $\delta$-function, the numerical error induced by source redistribution is local—mainly on the local mesh points to which the oxygen sources are distributed (see Appendix C). Nevertheless, the local oxygen field around the blood vessels must be sufficiently accurate for evaluating the oxygen flux using Eq (11). Therefore, without further improvement, we can only use a small spatial mesh size $h$ to perform simulations.

Next, we introduce a post-processing technique to reduce the local error induced by the redistribution of oxygen-sources. With this post-processing step, we are able to use a relatively large mesh size $h$ (e.g., be comparable to or even larger than vessel diameters) while maintaining a sufficiently small numerical error.

### 3.3 Post-processing

The post-processing is designed to reduce the error introduced by oxygen-source redistribution. This error can be defined as the difference $\delta P = P_O^{n+1} - \bar{P}_O^{n+1}$, where $P_O^{n+1}$ and $\bar{P}_O^{n+1}$ satisfy the following equations

$$\frac{1}{\Delta t}\left(P_O^{n+1} - P_O^n\right) = D\alpha\triangle_h P_O^{n+1} - M(P_O^{n+1}) + \sum_k q_k^{n+1}h_k\delta(\mathbf{x}_{\mathbf{k}} - \mathbf{x}),$$

$$\frac{1}{\Delta t}\left(\bar{P}_O^{n+1} - P_O^n\right) = D\alpha\triangle_h \bar{P}_O^{n+1} - M(\bar{P}_O^{n+1}) + \sum_k q_k^{n+1}h_k\bar{\delta}(\mathbf{x}_{\mathbf{k}} - \mathbf{x}),$$

respectively. Therefore, $\delta P$ satisfies

$$\frac{\delta P}{\Delta t} = D\alpha\triangle\,\delta P - \delta M + \sum_k q_k^{n+1}h_k(\delta(\mathbf{x}_{\mathbf{k}} - \mathbf{x}) - \bar{\delta}(\mathbf{x}_{\mathbf{k}} - \mathbf{x})), \qquad (16)$$

where $\delta M = M(P_O^{n+1}) - M(\bar{P}_O^{n+1}) \approx M'(P_O^{n+1})\delta P$.

Eq (16) appears to be nonlinear due to the nonlinearity in $\delta M$. However, in real applications, this term is always very small and we can neglect it. The reason that $\delta M$ is small is double fold: For mesh points near a vessel, the oxygen partial pressure is relatively high (much bigger than $P_0$), thus $M'(P_O^{n+1})$ is very small; whereas for mesh points away from all vessels, $\delta P$ becomes very small thanks to the good property of the numerical $\delta$-function. Therefore, we can evaluate $\delta P$ by neglecting the nonlinear term

$$\frac{\delta P}{\Delta t} \;=\; D\alpha \,\triangle\, \delta P + \sum_k q_k^{n+1} h_k (\delta(\mathbf{x_k} - \mathbf{x}) - \bar{\delta}(\mathbf{x_k} - \mathbf{x})). \tag{17}$$

Due to the linearity of Eq (17), $\delta P$ can be regarded as the linear combination $\delta P = \sum_k q_k^{n+1} h_k \delta P_k$, where $\delta P_k$ satisfies

$$\frac{\delta P_k}{\Delta t} \;=\; D\alpha \,\triangle\, \delta P_k + (\delta(\mathbf{x_k} - \mathbf{x}) - \bar{\delta}(\mathbf{x_k} - \mathbf{x})). \tag{18}$$

The boundary condition for Eq (18) is that $\delta P_k$ vanishes when $\mathbf{x} \to \infty$. In fact, as shown in Appendix C, only a few mesh-sizes away from $\mathbf{x}_k$, the error $\delta P_k$ is already negligible. This observation allows us to solve $\delta P_k$ only on local mesh near to $\mathbf{x_k}$ (e.g., an $8 \times 8$ mesh in the 2-D case).

The error $\delta P_k$ can be further decomposed into two parts $\delta P_k = P_k - \bar{P}_k$, where the first part $P_k = \int_0^{\Delta t} \frac{1}{\sqrt{(4\pi D\alpha s)^d}} \exp\left(-\frac{|\mathbf{x_k} - \mathbf{x}|^2}{4D\alpha s}\right) ds$ is the fundamental solution corresponding to the point source $\delta(\mathbf{x_k} - \mathbf{x})$ and the second part $\bar{P}_k$ is corresponding to the distributed sources

$$\frac{\bar{P}_k}{\Delta t} = D\alpha \,\triangle\, \bar{P}_k + \bar{\delta}(\mathbf{x_k} - \mathbf{x}), \tag{19}$$

$$\bar{P}_k\big|_{\partial\Omega_k} = P_k, \tag{20}$$

where $\partial\Omega_k$ is the boundary of the local domain $\Omega_k$. Note that the coefficient matrix to numerically solve the linear system (19) on the local mesh is independent of $k$ and the size of the coefficient matrix is small (e.g., $64 \times 64$). We can find and save the inverse of the coefficient matrix numerically. Then a matrix-vector multiplication is used to find $\bar{P}_k$.

In summary, for each source point $\mathbf{x_k}$, $\delta P_k$ is evaluated on a local mesh at the beginning. At each iteration step, the linear combination $\delta P = \sum_k q_k^{n+1} h_k \delta P_k$ is used to evaluate the error on mesh points close to vessels introduced by oxygen-source redistribution. This error is then removed from $\bar{P}_O^{n+1}$.

## 4 Parameters for simulation

The model parameters used in this work are listed in Table 1. The data are adapted from previous experiments on different animal retinas [41–45].

## 5 Numerical results

### 5.1 Blood vessel network structures

Four model structures of blood vessel networks are used in our simulation (see Fig 1). The simple cobweb structure (Fig 1(a)) in 2D and the single vessel (Fig 1(c)) in 3D are used for model validation and numerical convergence analysis. The 2D and 3D refined structures (Fig 1(b) and 1(d)) are adapted from real experimental measurements on a mouse retina. They are

**Table 1. Model parameters.**

| Blood oxygen parameters | | |
|---|---|---|
| Maximal RBC oxygen concentration | $C_0 = 0.5 cm^3 O_2/cm^3$ | [26] |
| Effective oxygen solubility | $\alpha_b = 3.1 \times 10^{-5} cm^3 O_2/cm^3/mmHg$ | [26] |
| Hill equation parameter | $P_{50} = 38 mmHg$ | [26] |
| Hill equation parameter | $n = 2$ | [26] |
| Tissue oxygen parameters | | |
| Diffusion constant | $D\alpha = 6 \times 10^{-10} cm^3 O_2/cm/s/mmHg$ | [26] |
| Consumption parameter | $M_0 = 1.7 \times 10^{-4} cm^3 O_2/cm^3/s$ | [26, 41] |
| Consumption parameter | $P_0 = 1 mmHg$ | [26] |
| Blood flow parameters | | |
| Oxygen partial pressure at inlet | $P_{b0} = 100 mmHg$ | [42] |
| Blood viscosity | $\mu = 4.6 \times 10^{-9} g/cm^2$ | [43] |

further used for analyzing the efficiency of oxygen supply. In this study, the no-flux boundary condition is used in solving the oxygen field with Eq (1).

## 5.2 Numerical solution for the cobweb vessel network

In Fig 2(a), we show the numerical solution of the oxygen partial pressure obtained with a $1024 \times 1024$ square mesh for the cobweb vessel structure. Since the distance between neighboring vessels is much bigger than that of real microvasculature, tissue away from the vessels is in a hypoxic state, i.e., the oxygen partial pressure is very low. The width of the well irrigated region for each vessel depends on the blood flow in it, being roughly on the scale of 100 $\mu m$. Note that the gap between neighboring capillaries in normal tissue is also roughly on the scale of 100 $\mu m$, which is much smaller than that in the cobweb structure. With such a high capillary density, normal tissue can be well irrigated.

The numerical error of the oxygen partial pressure is shown in Fig 2(b). The error is calculated by comparing the solution obtained with a $1024 \times 1024$ mesh and a $2048 \times 2048$ mesh. The error is smaller than 1 $mmHg$. This suggests that with a mesh size of about 5 $\mu m$, the numerical error can be smaller than one percent.

In Fig 3, we show the blood oxygen partial pressure $P_b$ and oxygen flux $q$ on all vessels. Because the blood vessels of the most inner loop are very small, they have a big resistance to blood flow. As a result, the blood flow in these vessels is very small, which leads to a fast decay in blood oxygen partial pressure along the flow direction (see Fig 3(a)). From Fig 3(c), we can see that the blood oxygen partial pressure $P_b$ decreases along each vessel segment as the oxygen diffuses from vessel to tissue. Similarly, in Fig 3(b) and 3(d), we show the oxygen flux $q$. From Fig 3(b), we can see that the flux $q$ reaches minimums at most cross-links of the vessel segments. This is because that more than one vessels share the oxygen supply to the local tissue at the cross-links. On the contrary, from Fig 3(d), we can see that the oxygen flux in vessel $a$ and $d$ reaches the maximums at their one or two ends. This is due to the vacancy of vessels at the center and the outer domain. At the end of the vessel segment $b$, the blood oxygen partial pressure becomes a constant and the flux becomes zero, because the oxygen partial pressure is higher in the surrounding tissue than the blood inside the vessel. Note that this unphysiological behavior is a consequence of the cobweb vessel structure. Here we set the negative flux to be zero. This setting does not significantly change the tissue oxygen level and blood oxygen concentration in downstream vessels since this piece of vessel is always short.

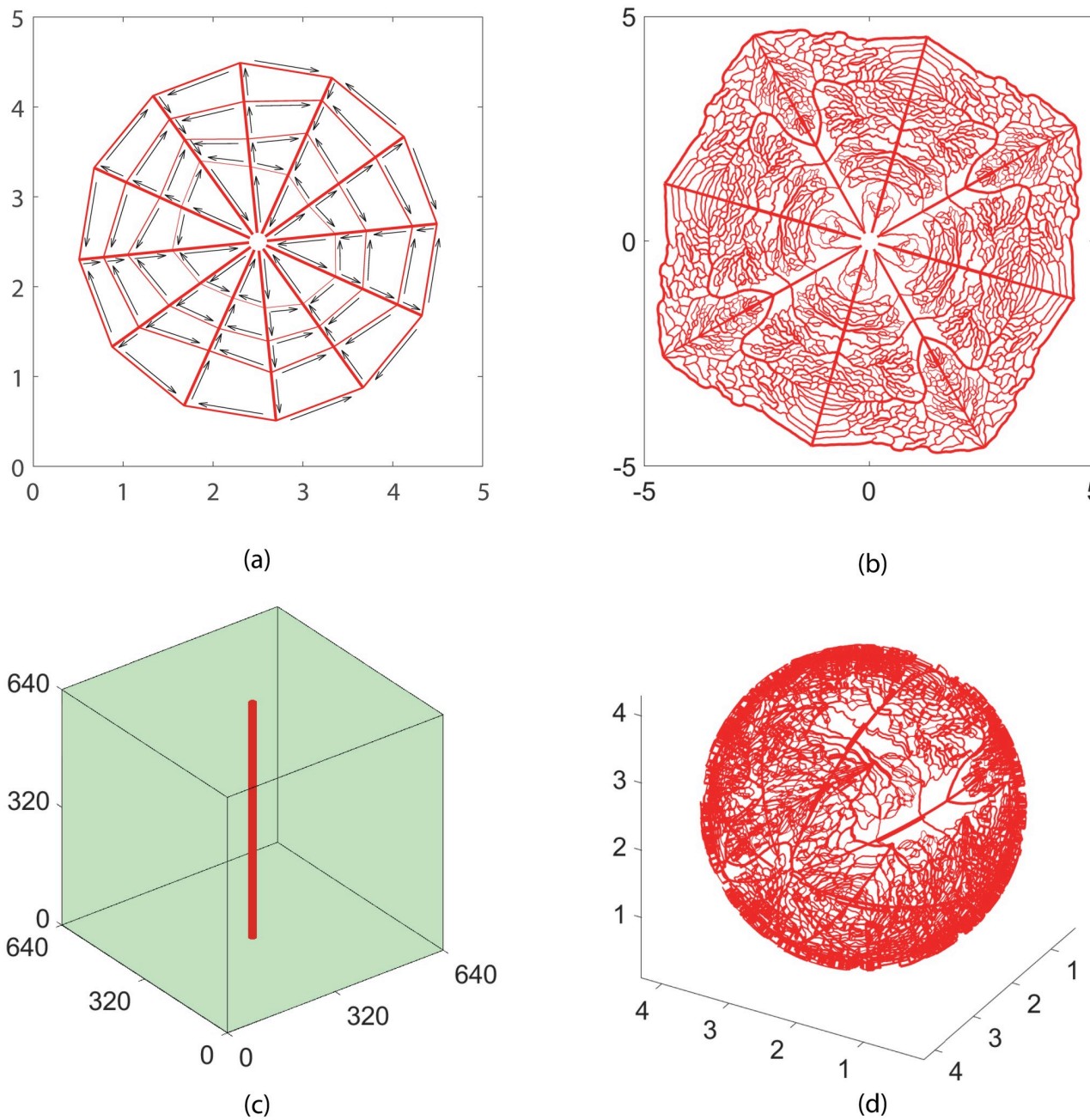

**Fig 1. Model vessel structures.** (a) A cobweb vessel structure mimicking the main branches of retinal vessel networks. There are six inlets and six outlets for blood flow near the center of the network. The inlets and outlets are in a spaced arrangement. The arrows show the direction of blood flow. (b) A refined vessel network with 4306 blood vessels. The network structure is adapted from a real retinal vessel network measured in experiment by stretching and symmetrical extensions. There are four flow inlets and flow outlets near the center. (c)A single vessel embedded in a tissue cube. (d) A refined 3D vessel network with 7815 blood vessels. The intermediate and deep capillary plexi layers in retina are reconstructed by projecting the above 2D network onto two spherical shells. The diameter of the two spheres for the deep capillary plexi layer, the intermediate layer, and the choroid layer is 2 *mm*, 2.1 *mm*, and 2.2 *mm*, respectively. The widths of the lines in each figure show the radii of blood vessels. The unit for the axes is micrometer for (c), and millimeter for others.

## 5.3 Numerical solution for the 2D refined vessel network

The numerical solutions of the oxygen partial pressure $P_O$ obtained with the refined vessel network with a $1024 \times 1024$ mesh are shown in Fig 4. The oxygen partial pressure field is obtained

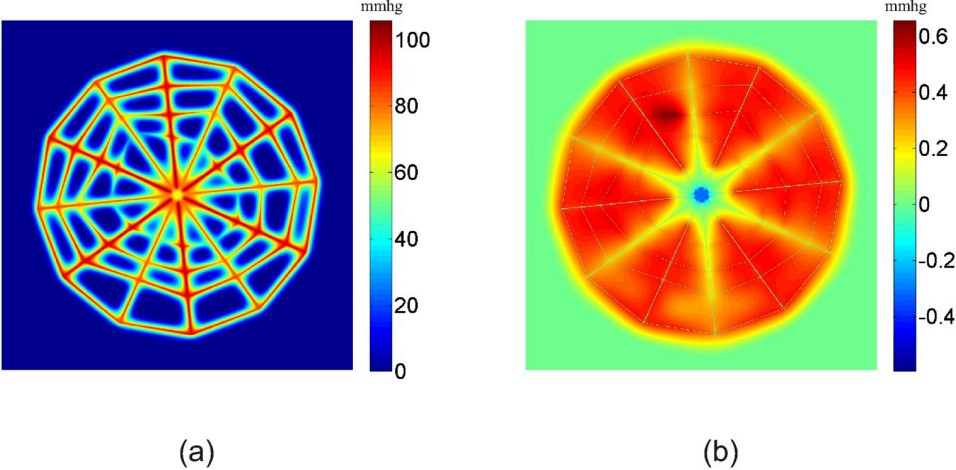

**Fig 2. Oxygen partial pressure and its numerical error.** (a) Numerical solution of the oxygen partial pressure obtained with a $1024 \times 1024$ square mesh. (b) Numerical error of the oxygen partial pressure calculated by the difference between the solution obtained with a $1024 \times 1024$ mesh and a $2048 \times 2048$ mesh. The unit for oxygen partial pressure is *mmHg*.

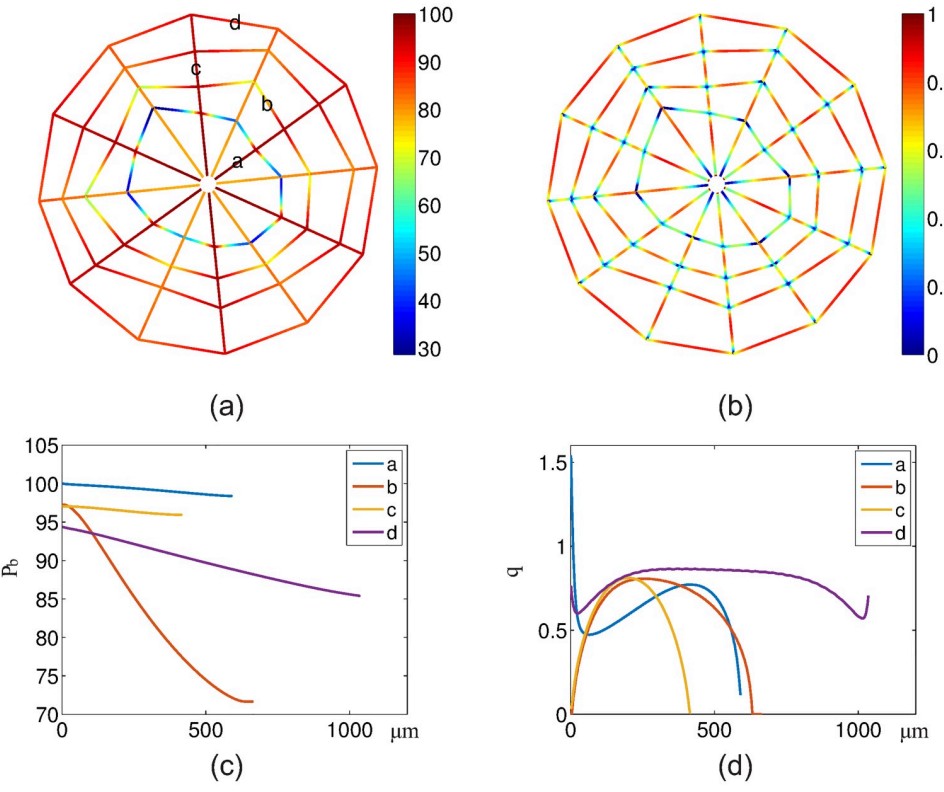

**Fig 3. Blood oxygen partial pressure and oxygen fluxes from the vessels.** (a) Blood oxygen partial pressure $P_b$ on the vessels. (b) Oxygen fluxes $q$ from the vessels to tissue. (c-d) $P_b$ and $q$ on four marked vessel segments in (a). The x-axis denotes the distance from the inlet for each vessel (i.e., arc-length coordinate). The unit of $P_b$ is *mmHg*. The unit of $q$ is $10^{-9} cm^3 O_2 / cm / s$.

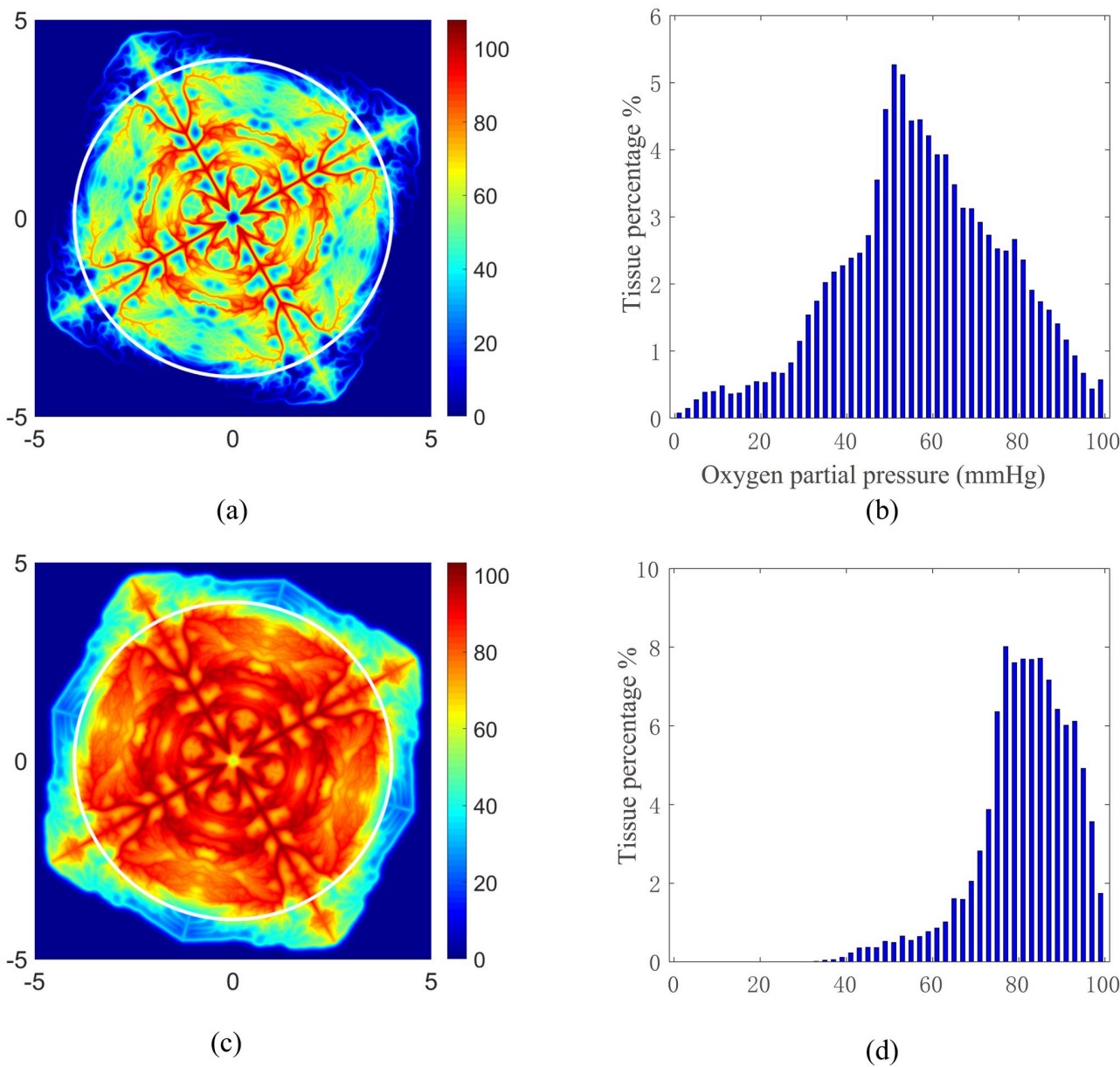

**Fig 4. Oxygen partial pressure obtained with the refined vessel network.** The partial pressure for normal tissue consumption and reduced tissue consumption are shown in (a) and (c), respectively. The tissue inside the white circles shown in (a) and (c) are used to statistically evaluate the area percentages of tissue with particular partial pressure of oxygen. The statistical results are shown in Figure (b) and (c), respectively. The unit of the $x-$ and $y-$ axes in (a) and (c) is *mm*. The unit of oxygen partial pressure is *mmHg*.

with a normal tissue consumption ($M_0 = 1.7 \times 10^{-4} cm^3 O_2/cm^3/s$) (see Fig 4(a)), whereas the partial pressure field is obtained with a reduced tissue consumption ($M_0 = 2.89 \times 10^{-5} cm^3 O_2/cm^3/s$) (see Fig 4(c)). In both cases, we can see that the oxygen partial pressure directly reflects the refined vessel structure. In Fig 4(b) and 4(d), we statistically analyze the oxygen fields inside the circles shown in Fig 4(a) and 4(c), respectively. The distance between the circle and the outer vessels is greater than 200 $\mu m$. Hence the boundary effects is insignificant. For the normal consumption case as shown in Fig 4(a) and 4(b), tissue in the outer domain has a very low oxygen supply. This is due to the large stretch in the outer domain when we generate the

refined vessel network, which significantly affects the oxygen supply in two folds: First, it increases the irrigated area of the corresponding blood vessels in the outer domain; Second, it increases the resistance of these vessels, thus decreases the blood flow in these vessels. For the reduced consumption case as shown in Fig 4(c) and 4(d), all tissue inside the circle is well irrigated. This statistical results of oxygen field under various tissue consumption are consistent with that reported in previous works [21, 46].

## 5.4 Numerical solution for the 3D refined vessel network

The 3D refined structure with 7815 vessels is adapted from the intermediate and deep capillary plexi of a mouse retina and used for simulation of the oxygen field with a $512 \times 512 \times 512$ mesh. The tissue consumption rate $M_0 = 2 \times 10^{-3} cm^3 O_2/cm^3/s$ and total blood flow $Q = 10 nl/s$ are adapted from experimental data to simulate the real situation [47, 48]. The simulated oxygen field at different layers is displayed in Fig 5. While low oxygen partial pressure can be observed in the layers away from the capillary plexus, the capillary-rich areas are well irrigated, which agrees well with the experimental data. Note that the irrigation efficiency depends on the total blood flow rate $Q$ and the vessel density.

## 5.5 Model validation

The comparison between our results and previous experimental and simulation results are shown in Fig 6. The horizontal axis represents the normalized retinal depth and the vertical axis represents the oxygen partial pressure. Despite the large differences in the independent experimental profiles, the profiles indeed share similar features, e.g., relatively high oxygen partial pressure is observed near the choroid surface and the deep capillary plexi, which is attributed to the high local vessel density. Meanwhile, a 'peak' can be observed in the middle, which corresponds to the intermediate capillary plexi layer. Despite the differences induced by particular parameter settings and experimental conditions, the simulation results are relatively consistent with the experimental data.

The comparison between our simulation results and that obtained with the numerical method developed in Ref. [26] are shown in Fig 7. Both simulations are performed on the same single-vessel domain as shown in Fig 1(C). The edge length of the cube is 640 $\mu m$ and the vessel radius is 10 $\mu m$. The oxygen profile in Fig 7(a) was obtained from a line through the cube center perpendicular to the vessel. All curves exhibit a similar diffusion distance that agrees well with the data under physiological conditions. Our results show good consistency with that obtained with the method developed in Ref. [26]. The blood oxygen partial pressure $P_b$ obtained with different mesh size is shown in Fig 7(b). A more detailed illustration of the error in oxygen field is shown in Fig 12. According to these results, a mesh size of $10\mu m$ is sufficient to achieve a relatively low numerical error (1%).

## 5.6 Convergence analysis and efficiency analysis

We use the cobweb vessel network in 2D and the simple single-vessel network in 3D to numerically study the convergence of our new method. The results for mesh refinement test are shown in Fig 8(a). The relative error $E_n$ is computed by the normalized $L^2$-norm of the difference between the solutions obtained with an $N^k$ mesh and a $2N^k$ mesh, where k is the dimension of the model. The numerical convergence order is about 1.7 for both the 2-D and 3-D cases. For the 3-D case, the relative error reaches at about 1% with a mesh size of $10\mu m$, which is consistent with the result shown in and Fig 7.

The iteration convergence is shown in Fig 8(b), where the relative difference is shown by the normalized $L^2$-norm of the difference of the corresponding functions, $q$ and $P_O$, between

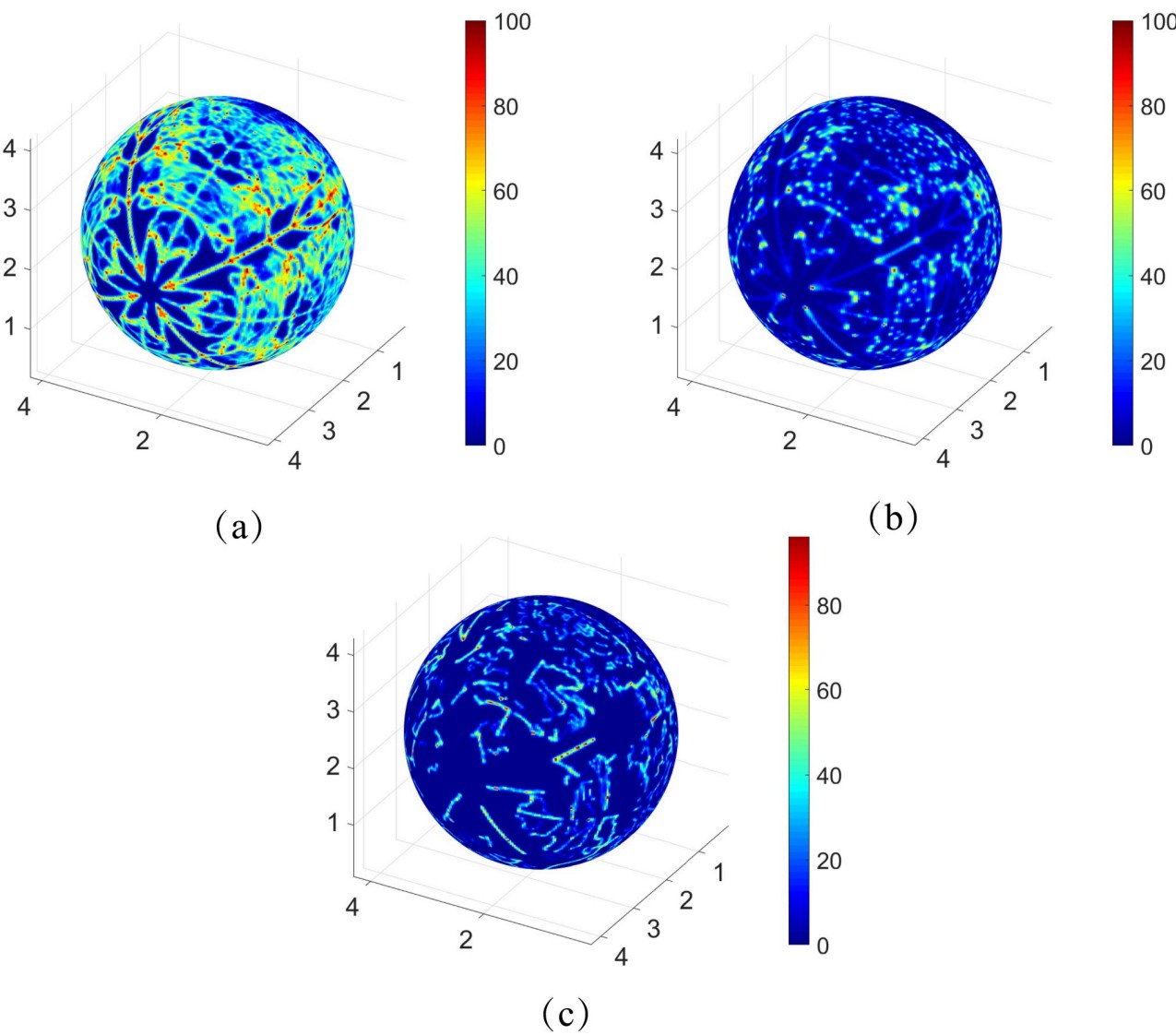

**Fig 5. The oxygen partial pressure at different layers of the retina.** The oxygen profiles on the layers corresponding to 0, 25, and 50 percent of the retina depth shown in the Figure (a), (b), and (c), respectively. The unit of the axes is mm and the unit of oxygen partial pressure is *mmHg*.

two iteration steps. We can see that the relative differences have a fast decay at the beginning, followed by a relatively slower linear convergence. In Fig 8(c), we show the iteration numbers required to make the relative difference smaller than $1 \times 10^{-3}$. We can see that the iteration numbers are comparable for the simple and refined vessel network structures, while both of them increase slightly with the mesh size $N$. The increase in iteration numbers is mainly due to the choice of the temporal step size $\Delta t$ in Eq (14). As we have discussed above, a large $\Delta t$ is helpful for convergence while a too large $\Delta t$ can induce instability in the iteration. According to our numerical test, the optimal $\Delta t$ decreases with $N$. For example, $\Delta t \approx 5000$ is used for $N = 1024$ in our simulation, while $\Delta t \approx 3500$ is used for $N = 2048$. The behavior in the 3-D case is similar.

The average number of Newton iterations required for each full iteration step is about $5 \sim 6$. Therefore, the time cost is about 40 seconds to find the solution with a relative error

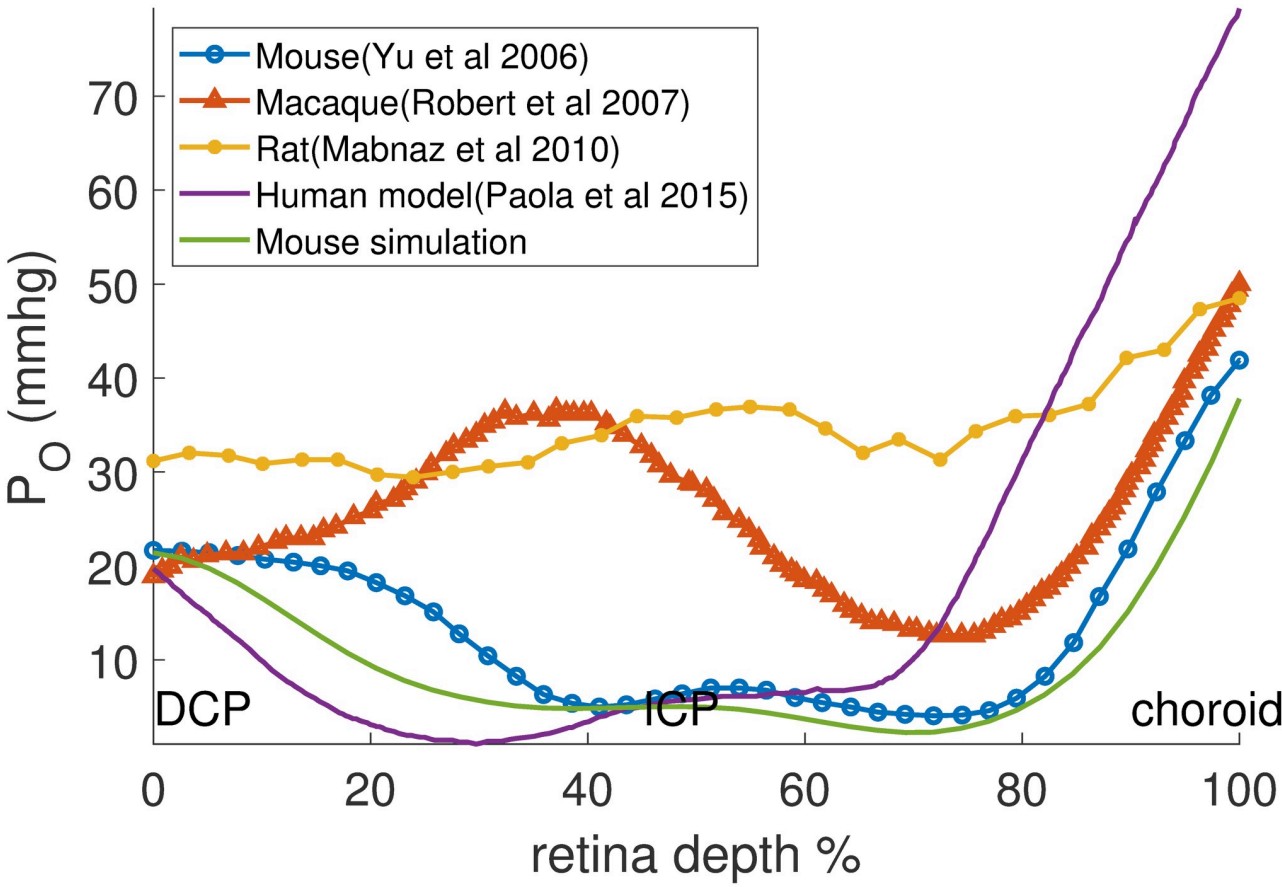

**Fig 6. Comparison between retina oxygen partial pressure profiles simulated by our model for the mouse retina and those experimentally measured in mice by [47], in macaques by [41], in rats by [48] and in a human retina model [31].** DCP: deep capillary plexi, ICP: intermediate capillary plexi.

smaller than 1% on a $1024 \times 1024$ mesh with a standard 3.7 GHz personal computer, whereas about 170 seconds on a $2048 \times 2048$ mesh. The iteration process for the 3-D simulation shares the similar convergence rate to that for the 2-D case. In our simulation, the complete 3-D simulation requires about 726 seconds on a $256 \times 256 \times 256$ mesh and about 8200 seconds on a $512 \times 512 \times 512$ mesh. This time increase from 2-D to 3-D comes mainly from the increase in solving the PDE.

The average time cost for each Newton iteration is analyzed in Fig 9. At the beginning of each iteration, three to six Newton iterations are required for each full iteration step. In each Newton iteration step, two to three V-cycles of multigrid iterations are required to solve the oxygen partial pressure field. After a few full iteration steps, only two to three Newton iteration steps, each with only one V-cycle, are sufficient to make the error small enough for each iteration step. In Fig 9(a) and 9(b), we show the average time cost for solving the oxygen field in tissue (*PDE-time*), solving the blood oxygen partial pressure (*ODE-time*), and post-processing (*P-time*) in each Newton iteration step, tested on a standard 3.7 GHz personal computer.

In our simulations, the step size along the blood vessels is set to be $h_k = \frac{h}{3} = \frac{L}{3n}$, where $L$ is the domain size and $n$ is the mesh size on one direction. For a given vessel network structure, the total number of discrete nodes on the blood vessels is proportional to $n$. Hence the sum of the ODE-time and P-time (*OP-time*) is also proportional to $n$. Note that for the standard

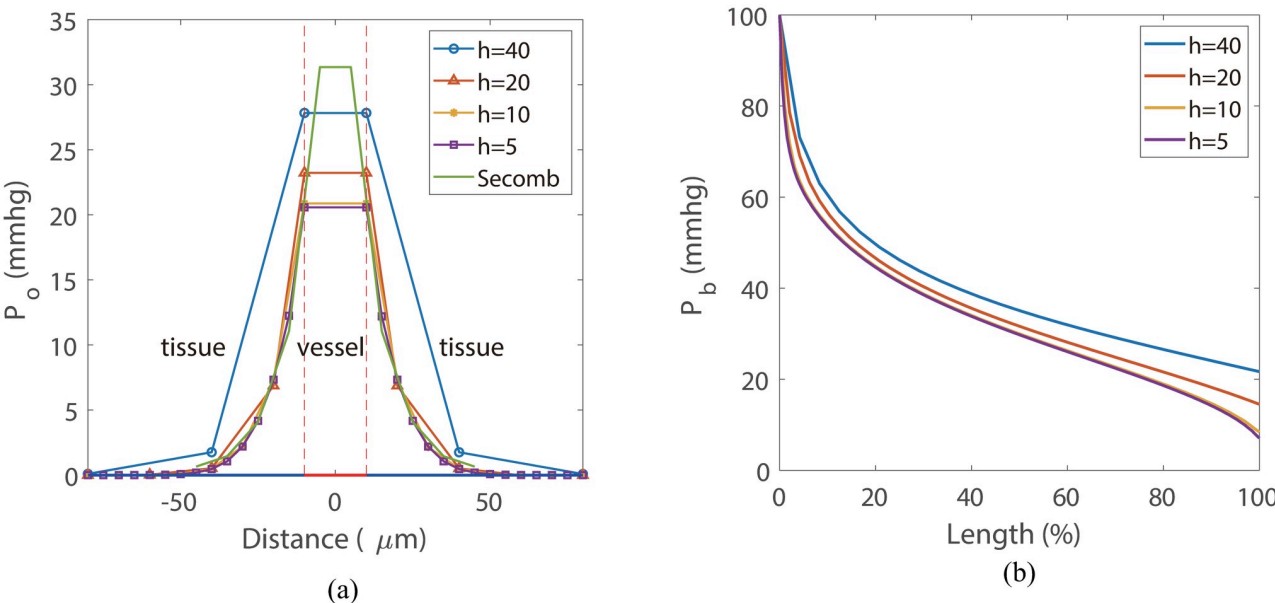

**Fig 7. Model validation and convergence analysis.** (a) The profiles of oxygen partial pressure on a line perpendicular to the vessel, where $x = 0$ represents cube center. The oxygen partial pressure inside the vessel $(-10, 10)$ are set to be equal to the blood oxygen partial pressure $P_b$. The line "Secomb" is obtained from the method developed in Ref. [26] with a mesh size of $10\mu m$. All simulations are performed with tissue consumption $M_0 = 2.0 \times 10^{-3} cm^3 O_2/cm^3/s$ and total blood inflow $Q_0 = 0.05 nl/s$. (b) The blood oxygen partial pressure $P_b$ along the blood vessel.

multigrid method, the computation complexity and the PDE-time are $O(n^2 \log n)$ for 2-D cases and $O(n^3 \log n)$ for 3-D cases, which increase much faster than the OP-time (see Fig 9(a) and 9(b)). As shown in Fig 9(c), for large mesh size $n$, the OP-time is only a small fraction of the PDE-time for the refined retina vessel networks.

In many real applications, simulations are required to perform in fully-vascularized 3-D tissue domains. Thus it is of interest to estimate the OP-time required in such cases. Obviously, the OP-time increases linearly with the total length of blood vessels. In principle, the OP-time is proportional to the total number of discrete grids on the blood vessels $N_v = \frac{L_v}{h_k}$, where $L_v$ is the total length of blood vessels. In order for a fair comparison for different simulations, we define the normalized time ration by

$$R_n = \frac{\text{OP-time}/N_v}{\text{PDE-time}/n^3} = \frac{\text{OP-time}}{\text{PDE-time}} \cdot \frac{n^3 h_k}{L_v}.$$

As shown in Fig 9(d), $R_n$ is relatively invariant with the mesh size $n$ and the total vessel length $L_v$. This suggests that the OP-time is really proportional to the total vessel length for given mesh size $n$.

For a fully vascularized tissue with fixed vessel density, the total vessel length is proportional to the volume $V$ of the tissue domain. As a result, for given mesh step size $h$, the ratio between the OP-time and the PDE-time is almost invariant with the domain size. Using the vessel density suggested in Ref. [33], the total vessel length in a cube with the edge length of 5.12 $mm$ is about 13422 $mm$, which is 13.3 times of that of the 3-D retina network (about 1012 $mm$) in this work. Therefore, for $h \leq 20\mu m$, the ratio between the OP-time and PDE-time is about smaller than 1 for fully vascularized tissue (see Fig 9(c)). In particular, for a fully vascularized $(5.12mm)^3$ cube, the OP-time is about 789 seconds for $h = 10\mu m$, and the total time cost is

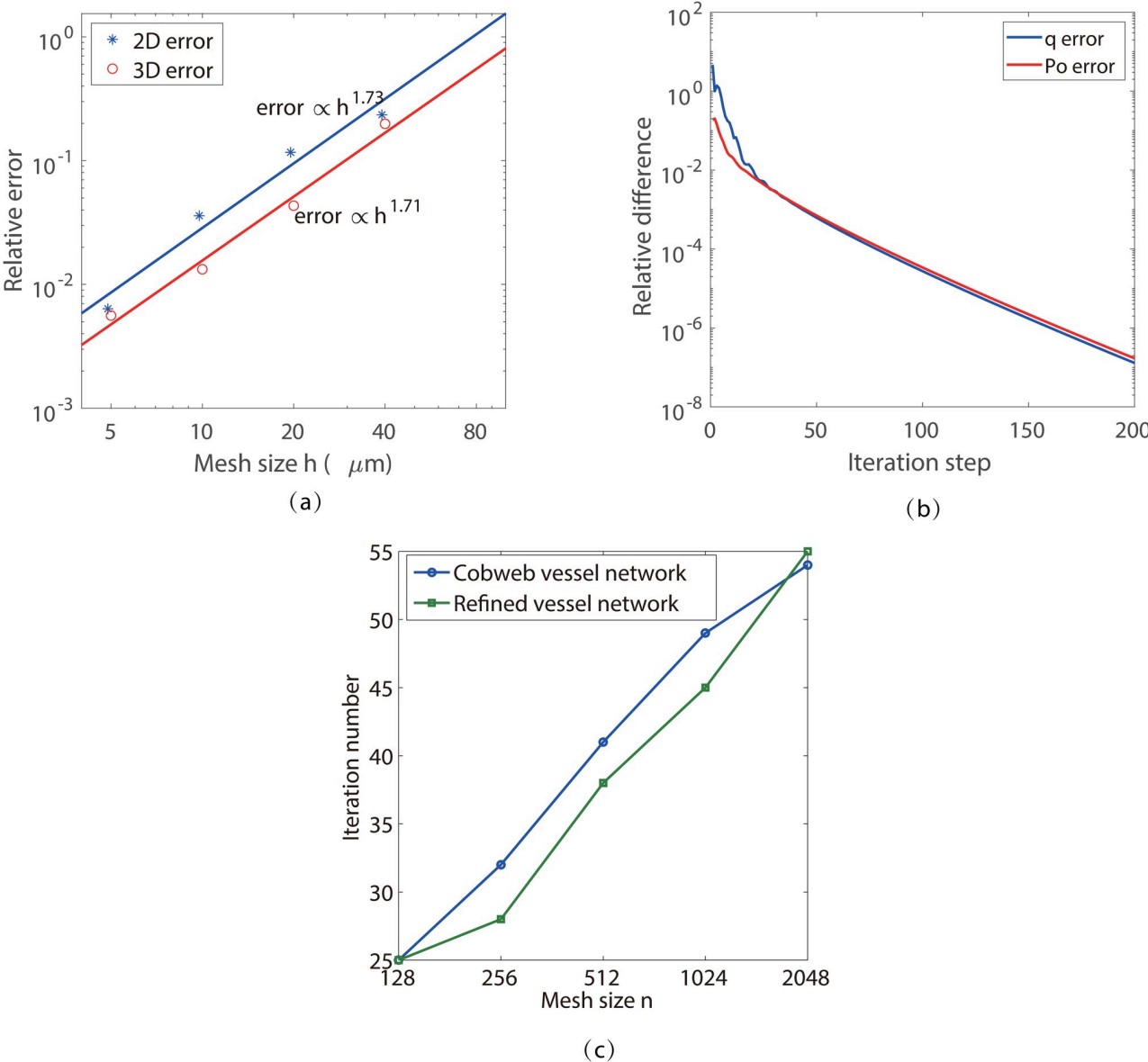

**Fig 8. Numerical convergence analysis.** (a)Mesh refinement test. The stars show the numerical error. The fitted lines show a convergence order of 1.73 for the 2D case and 1.71 for the 3D case with respect to mesh size. The relative error is computed by the normalized $L^2$-norm of $P_O$. (b) The decay of the relative difference between two iteration steps. The relative difference is computed by the normalized $L^2$-norm of $q$. (c) Iteration numbers for different mesh size.

about 8000 seconds; For a fully vascularized $(10.24mm)^3$ cube, which contains about $10^6$ vessel segments, the estimated total time is about 70390 seconds for $h = 10\mu m$.

## 6 Conclusions and discussions

Oxygen delivery in tissue plays an important role in many physiological processes such as angiogenesis, blood flow regulation, and blood vessel adaptation. Repeatedly evaluating the oxygen field in tissue is the key bottleneck that limits the large scale modeling and simulation of these important processes. In this work, a fast numerical method is developed for the

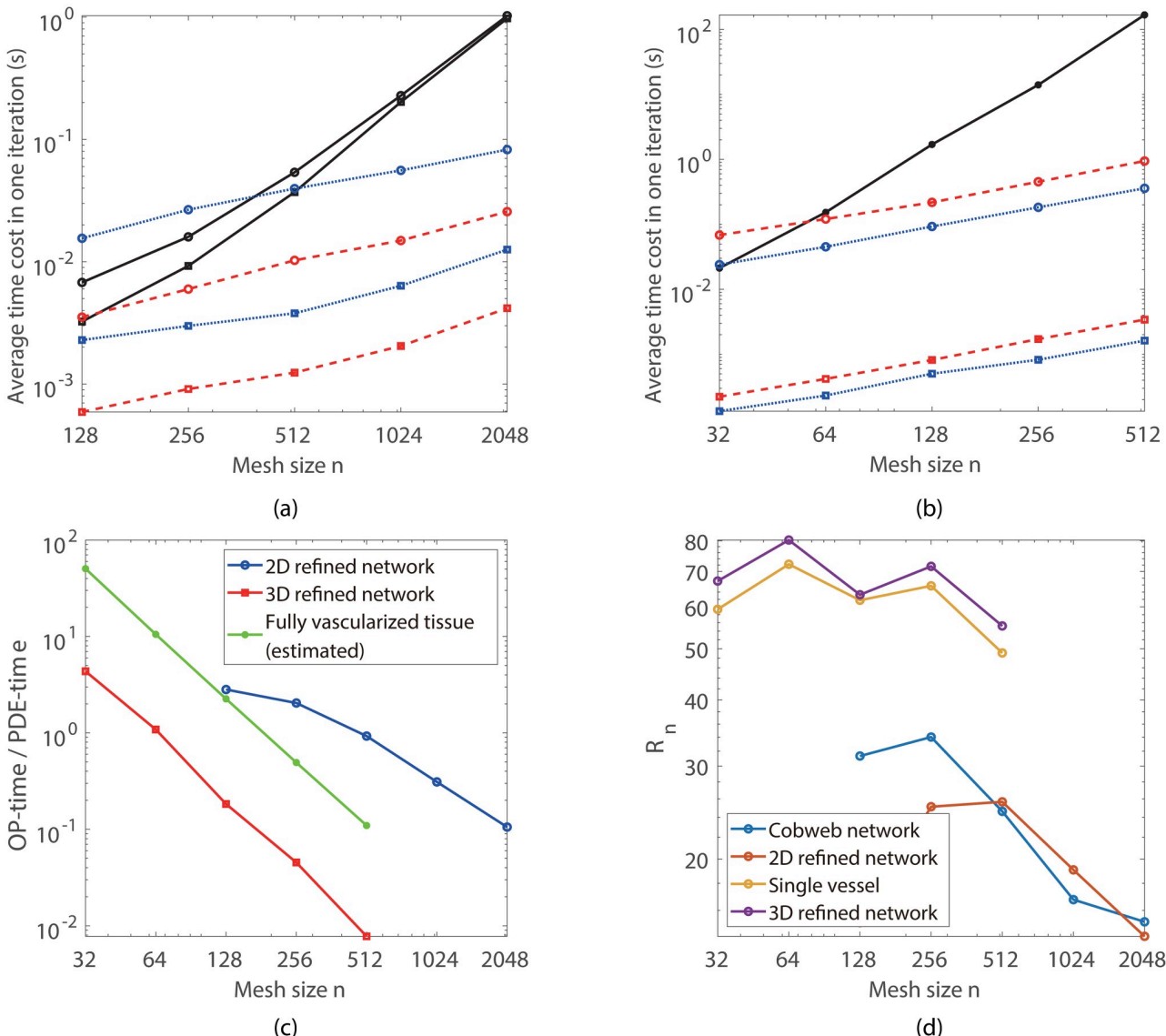

**Fig 9. Numerical efficiency analysis.** (a) and (b) Average time cost for each iteration step in the 2-D and 3-D cases, respectively. The black solid lines, the blue dotted lines, and the red dashed lines show the average time cost for solving the PDE, the ODEs, and the post-processing in each Newton iteration step, respectively. For the 2-D case, the circled and squared lines show the time cost for the refined vessel network and the simple cobweb network, respectively; whereas for the 3-D case, the circled and squared lines show the time cost for the refined retina network and the single vessel system, respectively. (c) The time ratio between the OP-time and PDE-time for the 2-D and 3-D cases. (d) The normalized time ratio for different systems.

computation of the nonlinear coupled system of oxygen consumption, oxygen diffusion, and oxygen delivery in blood vessels with complex network structures.

Our fast numerical method involves an implicit finite-difference method for solving the partial differential equation of oxygen partial pressure with a square mesh. The key techniques we used include (1) the Peskin's numerical $\delta$-function to distribute the oxygen sources onto mesh points and (2) the post-processing to remove the numerical error induced by the distribution of oxygen sources. With these techniques, relatively large spacial mesh size can be used while sufficient numerical accuracy is maintained. The computational complexity is slightly bigger than linear with respect to the number of mesh points, taking into account the increase

in iteration steps for refined meshes. The convergence of numerical error is slightly less than second order with respect to the mesh size. For three-dimensional simulations, the numerical error can be controlled to be about 1% with a mesh step size of $h = 10\mu m$.

Although we have not performed a 3-D simulation of fully vascularized tissue because we do not have a suitable 3-D vascular structure, our numerical tests show that for given step size $h \leq 20\mu m$, the simulation time is mainly costed in the multigrid method for solving the oxygen field. A large scale simulation in a fully vascularized $(10.24mm)^3$ cube can be achieved within 20h for $h = 10\mu m$. Moreover, the natural extension of our method from serial to parallel also leaves abundant possibilities for further applications on various large organs.

Nevertheless, we can see that the total number of iterations used in our current simulations is still large. Hence, to pave the way for more realistic three-dimensional simulations with complex blood vessel networks, better iteration strategies should be explored to reduce the time cost in our method.

## 7 Appendix

### A Oxygen flux through blood vessel walls

For the two-dimensional case, oxygen flux is evaluated on both side of the vessel walls separately. The oxygen partial pressure $P_O$ is assumed to be equal to $P_b$ on vessel walls. In order to evaluate the oxygen flux at a discrete node $\mathbf{x} = (x, y)$ on the vessel center line, we first numerically calculate the unit normal direction $\mathbf{n}$ by central difference. Then, we define two corresponding points $\mathbf{x}^{\pm} = \mathbf{x} \pm r\,\mathbf{n}$ on the vessel wall (see Fig 10), where $r$ is the vessel radius. Next, the oxygen partial pressures $P_O$ on four nearest mesh points to $\mathbf{x}^+$ outside the vessel are used to fit a linear function $P_O(x, y) \approx P_b + \mathbf{G} \cdot (\mathbf{x} - \mathbf{x}^+)$ in the sense of least squares, where $\mathbf{G}$ is the gradient to be fitted. Finally, the oxygen flux is given by $q = D\alpha\,\mathbf{G} \cdot \mathbf{n}$.

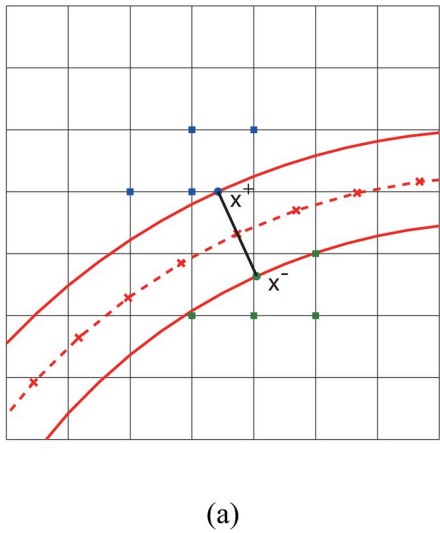
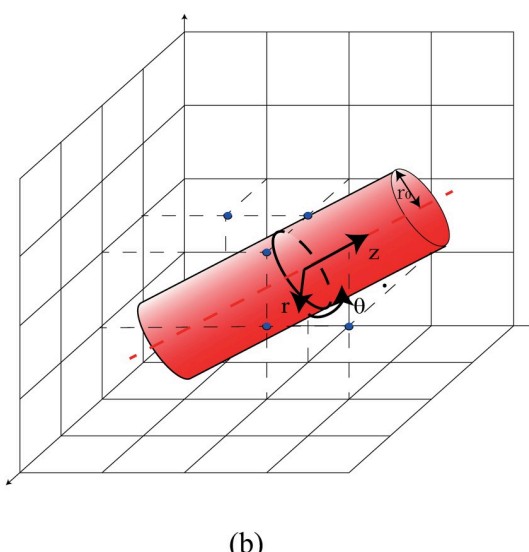

(a) (b)

**Fig 10.** (a) Evaluation of oxygen flux in 2D. The red solid lines represent the vessel wall. The red dashed line shows the center line of the blood vessel, the crosses on which show the discrete nodes. The blue (green) solid circle shows $x^+$ ($x^-$), whereas the blue (green) solid squares show the nearest mesh points used to fit the linear function and evaluate the flux. (b) Evaluation of oxygen flux in 3D. The red dashed line shows the center line of the blood vessel. The black circle shows the cross-section of blood vessel, whereas the blue (green) solid squares show the nearest mesh points used to fit the solution and evaluate the flux.

For the three-dimensional case, the oxygen field near the blood vessel can be described by the Laplace equation under cylindrical coordinate system.

$$\frac{1}{r}\frac{\partial}{\partial r}\left(r\frac{\partial P_O}{\partial r}\right) + \frac{1}{r^2}\frac{\partial^2 P_O}{\partial \theta^2} + \frac{\partial^2 P_O}{\partial z^2} = 0,$$

$$P_O(r_0, \theta, z_0) = P_b,$$

where $r_0$ denotes the radius of the blood vessel, $z_0$ and $P_b$ are z coordinate of one discrete node on vessel center line and the corresponding blood oxygen partial pressure, respectively. Then the series expansion of the solution can be writen as,

$$\begin{aligned}
P_O(r, \theta, z) &= (a_1 + a_2 z)(a_3 + a_4 \log r) \\
&+ (a_5 + a_6 z)(a_7 r + a_8/r)(a_9 \cos\theta + a_1 0 \sin\theta) + \cdots
\end{aligned}$$

By keeping the first term and applying the boundary condition, the solution can be written as

$$P_O(r, \theta, z) = P_b + c_1 \log(r/r_0) + c_2(z - z_0) + c_3(z - z_0)\log(r/r_0).$$

In our numerical tests, a few nearest mesh points are used fit the function in the sense of least squares. The distance between the fitting point and the point on center line is less than $r_0 + h$. Finally, the oxygen flux is obtained by $q = 2\pi D\alpha c_1$.

## B Peskin's numerical $\delta$-function

Peskin's numerical $\delta$-function is defined as

$$\bar{\delta}(\mathbf{x}) = \frac{1}{h^d}\prod_{i=1}^{d}\phi(\frac{x_i}{h}),$$

where $d$ is the dimension of $\mathbf{x}$, h is the mesh size, and $x_i$ are the Cartesian components of $\mathbf{x}$. The one-dimensional continuous function $\phi(x)$ is given by

$$\phi(r) = \begin{cases}
0, & r \leq -2; \\
\frac{1}{8}(5 + 2r - \sqrt{-7 - 12r - 4r^2}), & -2 \leq r \leq -1; \\
\frac{1}{8}(3 + 2r + \sqrt{1 - 4r - 4r^2}), & -1 \leq r \leq 0; \\
\frac{1}{8}(3 - 2r + \sqrt{1 + 4r - 4r^2}), & 0 \leq r \leq 1; \\
\frac{1}{8}(5 - 2r - \sqrt{-7 + 12r - 4r^2}), & 1 \leq r \leq 2; \\
0, & r \geq 2.
\end{cases}$$

$\phi(x)$ satisfies the following constraints

$$\begin{cases} \phi(x) = 0 \ \text{ for } \ |x| > 2; \\[2mm] \sum_{j \ even} \phi(x-j) = \sum_{j \ odd} \phi(x-j) = \frac{1}{2} \ \text{ for all real } \ x; \\[2mm] \sum_j (x-j)\phi(x-j) = 0 \ \text{ for all real } \ x; \\[2mm] \sum_j \left(\phi(x-j)\right)^2 = const \ \text{ for all real } \ x. \end{cases}$$

## C Localization of $\delta P_k$

An example of $\delta P_k$ defined in Eq (18) on the local mesh is illustrated in Fig 11, where the mesh size is $h = 0.002$, $\mathbf{x_k} = (0.5 + h/2, 0.5 + h/2)$ is set at the center of the local mesh. It can be clearly seen that $\delta P_k$ decays very rapidly. When $|\mathbf{x} - \mathbf{x_k}|>2h$, the error is already negligible.

## D Detail error analysis for single vessel

A detailed error analysis for this classical Krogh model is shown in Fig 12. In order to estimate the numerical error induced by the spatial discretization, we performed simulations with mesh sizes of 5,10,20 and 40 $\mu m$, respectively. The result of 5-$\mu m$ mesh was used as a reference to determine the errors of those obtained with other mesh sizes. It can be observed that the errors are mainly distributed in the vessel and surrounding tissue. A space step of 40 $\mu m$ will bring

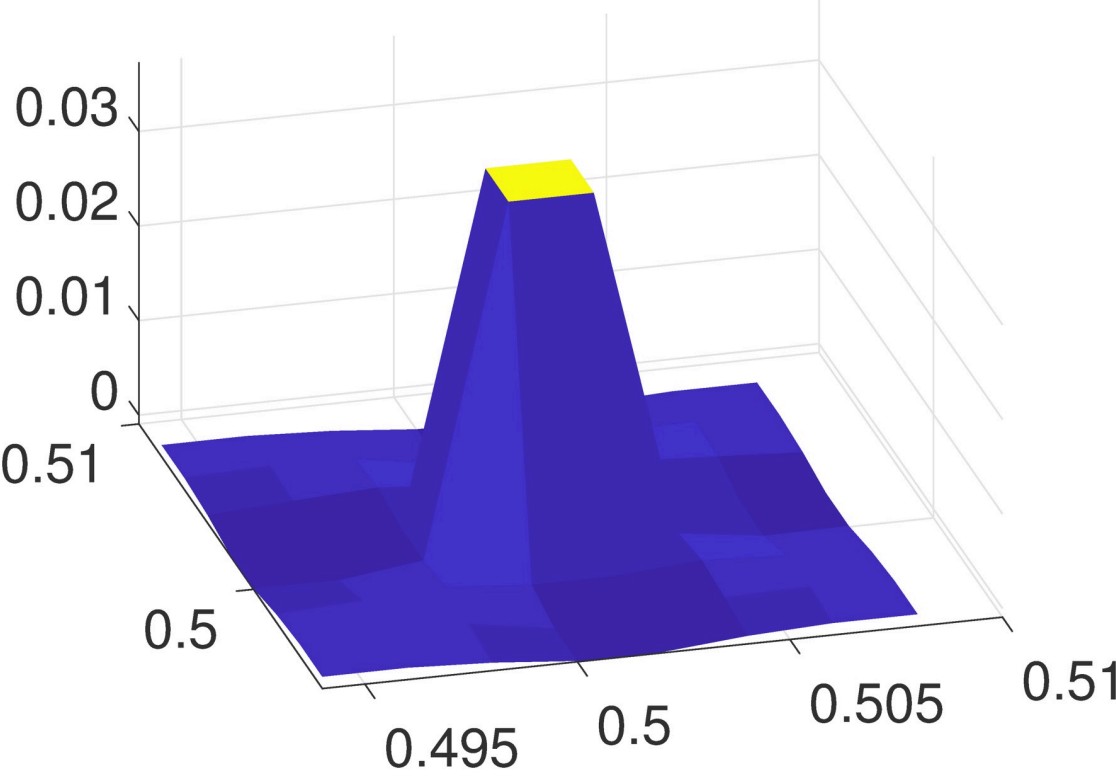

**Fig 11. The relative error $\delta P_k$ on a local mesh.**

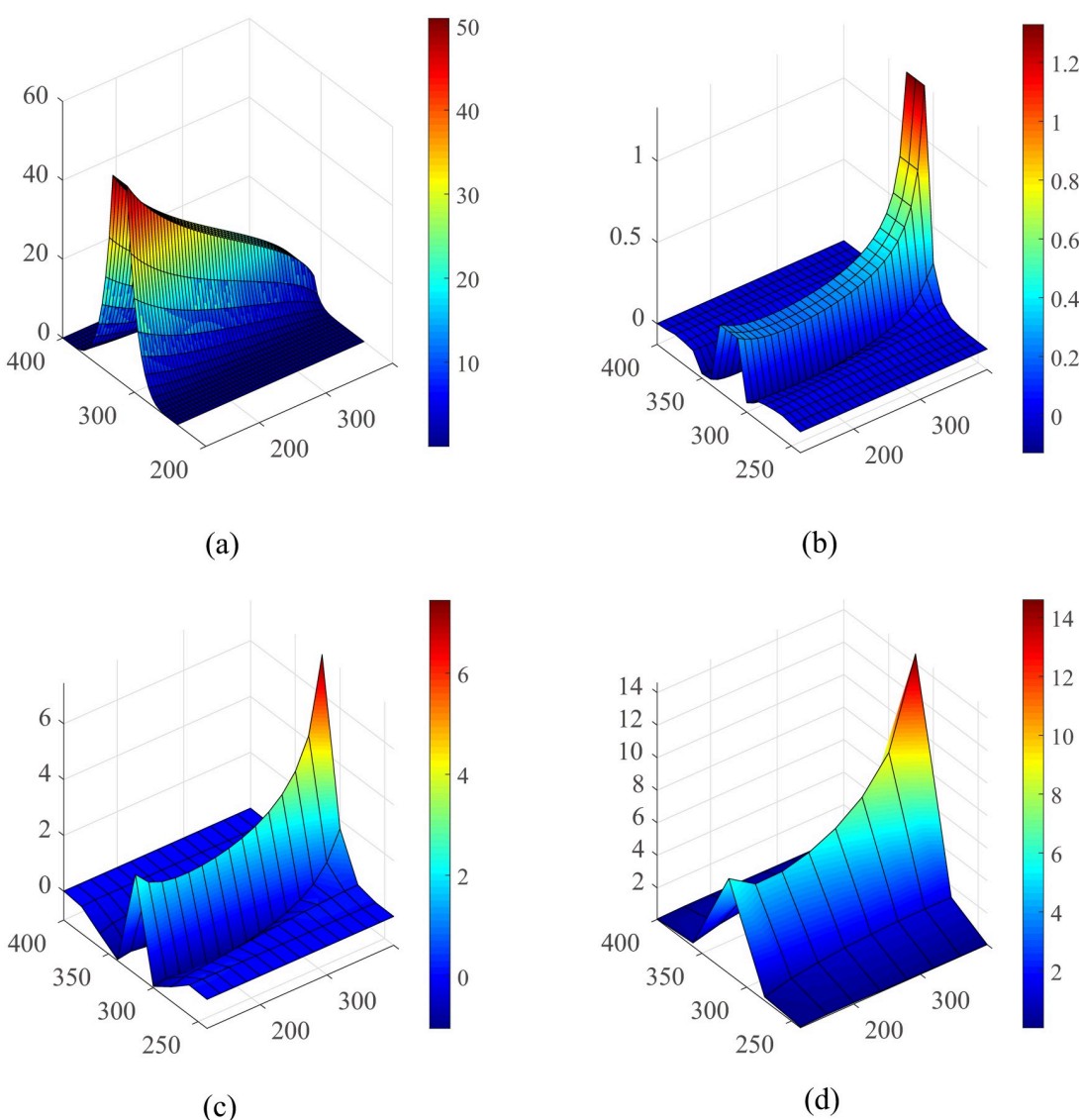

**Fig 12. Detailed oxygen field error analysis for the single vessel model.** (a) The oxygen profile on the plane of the vessel under the fine grid ($h = 5\mu m$). The numerical error for the $h = 10$, $h = 20$ and $h = 40$ are shown in (b-d), respectively, while the result of 5-$\mu m$ mesh were used as a standard. The unit of the axes is $\mu m$ and the unit of oxygen partial pressure is $mmHg$.

**Table 2.**

| vascular system | Number of inlets | inlet flow | Number of outlets | outlet flow |
|---|---|---|---|---|
| 2D cobweb network | 6 | 0.6 nl/s | 6 | 0.6 nl/s |
| 2D refined network | 4 | 1 nl/s | 4 | 1 nl/s |
| 3D single vessel | 1 | 0.05 nl/s | 1 | 0.05 nl/s |
| 3D refined network | 4 | 2.5 nl/s | 4 | 2.5 nl/s |

about 10% numerical error in the blood vessel, where the case of 10 $\mu m$ only has an error less than 1%.

### E Inflow and outflow conditions

The inflow and outflow conditions of the four vascular systems used in in this work are listed in Table 2. When there are multiple inlets and outlets, they share the same blood flow rate.

## Supporting information

**S1 File.**
(ZIP)

## Author Contributions

**Conceptualization:** Dan Hu.

**Methodology:** Yuankai Lu, Dan Hu.

**Resources:** Wenjun Ying.

**Software:** Yuankai Lu, Wenjun Ying.

**Supervision:** Dan Hu.

**Writing – original draft:** Yuankai Lu.

**Writing – review & editing:** Yuankai Lu, Dan Hu.

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
