## [Decision Letter · Decision Letter 0]

20 Jan 2021

PONE-D-20-33706

A fast numerical method for oxygen supply in tissue with complex blood vessel network

PLOS ONE

Dear Dr. Hu,

Thank you for submitting your manuscript to PLOS ONE. After careful consideration, we feel that it has merit but does not fully meet PLOS ONE’s publication criteria as it currently stands. Therefore, we invite you to submit a revised version of the manuscript that addresses the points raised during the review process.

A rebuttal letter that responds to each point raised by the reviewer(s). You should upload this letter as a separate file labeled 'Response to Reviewers'.A marked-up copy of your manuscript that highlights changes made to the original version. You should upload this as a separate file labeled 'Revised Manuscript with Track Changes'.An unmarked version of your revised paper without tracked changes. You should upload this as a separate file labeled 'Manuscript'.

We look forward to receiving your revised manuscript.

Kind regards,

Adélia Sequeira, Ph.D

Academic Editor

PLOS ONE

Additional Editor Comments:

The manuscript still needs some improvement according to the comments made by the reviewers.

Please do that and resubmit the paper.

"This work is supported by the National Natural Science Foundation of China (Contract 433

no. 11971312, 11771290, and 91630208) and Student Innovation Center, Shanghai Jiao 434

Tong University."

"This work is supported by the National Natural Science Foundation of China (Contract no. 11971312, 11771290, and 91630208). The funders had no role in study design, data collection and analysis, decision to publish, or preparation of the manuscript."

<h1>** **</h1>

Reviewers' comments:

Reviewer's Responses to Questions

**Comments to the Author**

1. Is the manuscript technically sound, and do the data support the conclusions?

Reviewer #1: Yes

Reviewer #2: Yes

2. Has the statistical analysis been performed appropriately and rigorously? 

Reviewer #1: N/A

Reviewer #2: Yes

3. Have the authors made all data underlying the findings in their manuscript fully available?

Reviewer #1: Yes

Reviewer #2: Yes

4. Is the manuscript presented in an intelligible fashion and written in standard English?

Reviewer #1: Yes

Reviewer #2: No

5. Review Comments to the Author

Reviewer #1: The authors consider combined model of microvascular flow and oxygen transport in the surrounding tissue.

I think, that the authors provided proper feedback to the previous reviewer comments and the text looks much better although I didn't find what was highlighted in green color. I suggest, that the 3D fully vascularised case should be a matter of separate study. The author comments in this regard are enough.

Nevertheless I have several issues, which should be addressed before final approval.

1) The purpose of the oversimplified test examples is not clear (Fig 1 a, c). These cases may be removed or more details should be given (e.g. comparison with analythics, with some other standrad solvers, etc). In the current version it seems thay are not used for the model validation or it is not decribed in the text.

2) Lines 283-284: The conditions are unphysiological (oxygen partial pressure in the tissure greater thna in the microvessels). It should be some how commented.

3) Dimensions must be present on all graphs! (mmHg, percents, seconds, etc)

4) What are the inputs and outputs for the networks? What are the boundary conditions?

5) Why different curves finished at different valu along x axis? (see Fig. 3 c, d)

6) Lines 292-302: Statistical analysis must be improved. It is obvous, that boundary parts introduce significant error. They are not exist in reality, so they are unphysiological, but the model needs them as, say, variant of boundary conditions. I suggest to perform analysis separately in the dense area (with physiological density of capillary network). This region must demonstrate good correlation with experiments.

7) How the length, diamters and hydraulic resistance were set to all structures?

8) The authors should consider these recent works, which closely relate to the review and methods sections: DOI: 10.1134/S199508021605005X, 10.1134/S0006350916050183, 10.20537/2076-7633-2017-9-3-487-501, 10.3390/MATH8081204, 10.3390/MATH8050760, 10.1134/S0006350919020118, 10.1515/rnam-2018-0019.

Reviewer #2: Essentially the paper is well written and organized, after the first revision. All in all, I can recommend a minor revision of the submitted paper. My specific comments are summarized in the attached PDF. See items 1-10.

6. PLOS authors have the option to publish the peer review history of their article (what does this mean?). If published, this will include your full peer review and any attached files.

Reviewer #1: No

Reviewer #2: No

---

## [Author Response · Author response to Decision Letter 0]

9 Feb 2021

We preciate the important and thoughtful comments raised by the reviewers. Their comments and suggestions are very helpful for us to improve our work. We have modified the manuscript accordingly. In the manuscript, the modified sentences are highlighted in green.

Reply to the first reviewer:

1.1 The purpose of the oversimplified test examples is not clear (Fig 1 a, c). These cases may be removed or more details should be given (e.g. comparison with analythics, with some other standard solvers, etc). In the current version it seems thay are not used for the model validation or it is not decribed in the text.

Due to the complicated coupling of blood flow and tissue oxygen supply, as well as the nonlinearity in the system, it is difficult to obtain an analytical solution. The oversimplified test examples are used to validate our model and algorithm as that in previous studies [33,36]. 

The 2-dimensional cobweb structure allows us to qualitatively analyze the behavior of the model and our numerical method. For example, the width of the well irrigated region can be clearly seen from Figure 2(a) (Line 294-298). From Figure 2(b), the readers can also get clear sense on the distribution of numerical error (Line 302-305). The detailed behavior of the oxygen flux q and blood oxygen partial pressure Pb along blood vessels are shown in Figure 3 and discussed on Line 306-320. Note that the complex geometry of the refined structure can bring difficulties in these discussions and the over simplified is helpful in this sense.

Furthermore, the cobweb structure is also used to compare the convergence efficiency of the nonlinear iteration (Figure 8(c)). We can see that the iteration numbers are comparable for the cobweb structure and the refined vessel network structures (Line 389-390).

The three dimensional single-vessel structure is used to analyze the numerical efficiency of our method and show the consistency of our numerical solution with that obtained with a widely accepted numerical algorithm [26].

From these points of view, we feel that studies based on the oversimplified examples (Figure 1 a, c) are helpful for validation of both the model and our numerical method. We understand that such oversimplified structures may bring misleading information on the solutions. We have added emphases to avoid such misleading information (Line 294-297 ).

1.2 Lines 283-284: The conditions are unphysiological (oxygen partial pressure in the tissure greater than in the microvessels). It should be some how commented. 

The reviewer is right. This result is physically correct but physiologically unreasonable due to the unphysilogical setup of the cobweb geometry which contains a few sharp bifurcation angles. We have emphasized that the geometry and the consequence phenomena are unphysilogical on Line 316-317.

1.3 Dimensions must be present on all graphs! (mmHg, percents, seconds, etc)

We have revised all graphs accordingly. The descriptions of the unit are added in the captions of Figure 3, Figure 4, Figure 5, and Figure 12 in the manuscript.

1.4 What are the inputs and outputs for the networks? What are the boundary conditions?

The input and output flow conditions for the network are now collected in Appendix E case by case now. The no-flux boundary condition for Equation (1) are used in this work (information added on Line 285-286).

1.5 Why different curves finished at different value along x axis? (see Fig. 3 c, d)

As noted in the caption of Fig 3 , the x-axis denotes the distance from the inlet of each vessel (i.e., arc-length coordinate). The four blood vessels have different lengths, hence the curves finished at different value along x axis. 

1.6 Lines 292-302: Statistical analysis must be improved. It is obvous, that boundary parts introduce significant error. They are not exist in reality, so they are unphysiological, but the model needs them as, say, variant of boundary conditions. I suggest to perform analysis separately in the dense area (with physiological density of capillary network). This region must demonstrate good correlation with experiments.

The reviewer is right that our previous statistical analysis is polluted by the boundry effects. We have performed the analysis again in a smaller domain to avoid the boundary effects. The new results are now included in the revised Figure 4. The statistics of oxygen field under various tissue consumption are also discussed in previous works [21,45], and show the similar features. Corresponding comments are added on Line 328-330, 337-338, Page 11.

1.7 How the length, diamters and hydraulic resistance were set to all structures?

We have added an Supplementary file to include all the geometrical data. The hydraulic conductance (the reciprocal of the resistance) are obtained using the Poiseuille law (Equation (3) ). Note that it is also possible to incorporate and radius-dependent viscousity in this equation. 

1.8 The authors should consider these recent works, which closely relate to the review and methods sections: DOI: 10.1134/S199508021605005X, 10.1134/S0006350916050183, 10.20537/2076-7633-2017-9-3-487-501, 10.3390/MATH8081204, 10.3390/MATH8050760, 10.1134/S0006350919020118, 10.1515/rnam-2018-0019.

These references are closely related to our new numerical method and provides potential applications of our new method in the angiogenic tumor growth, progression, and therapy. We have added these references accordingly (Line 12-14).

Reply to the second reviewer:

2.1 The authors motivate the objective of their work by means of angiogenesis. However, there is another application area for models that can be used to simulate oxygen supply of tissue: Generation of artifical microvascular networks. To obtain further information on this topic one could consider e.g. the following publications: { T. K¨oppl, E. Vidotto & B. Wohlmuth (2020). A 3D-1D coupled blood flow and oxygen transport model to generate microvascular networks. International Journal for Numerical Methods in Biomedical Engineering, DOI: 10.1002/cnm.3386 { M. Schneider, J. Reichold, B. Weber, G. Szekely & S. Hirsch (2012). Tissue metabolism driven arterial tree generation. Medical image analysis, 16(7), 1397-1414.

The reviewer’s suggestion are very helpful for us. We have added these literatures (Line 14-17) and discussed the potential application of our method in generation of artificial microvascular networks. 

2.2 In Section 2.2., blood flow and blood pressure are modeled by means of the Poiseuille model without taking the exchange with the surrounding tissue into account using e.g. Starling’s law. Please justify, why the fluid exchange through the porous vessel wall is omitted.

There is indeed fluid exchange through the porous vessel wall. This exchange may only be a small percent of the total flow (e.g. 0.5%, see Ref [35] ) and the Poiseuille model is accurate. When this exchange becomes large, such as in the kidney or in particular pathological states, the Starling’s law can be used by introducing flow sinks in Equation (4). We have added these discussions and corresponding references on Line 98-103.

2.3 In Section 2.4, the authors model the oxygen exchange between the tissue and the vascular system by means of the averaged PO2 gradient in 3D. However in several publications e.g.: L. Cattaneo & P. Zunino (2014). A computational model of drug delivery through microcirculation to compare different tumor treatments. International journal for numerical methods in biomedical engineering, 30(11), 1347-1371. the exchange of substances like oxygen between the vascular system and tissue is modeled by the Kedem-Katalchsky law, which is a standard filtration law for permeable membranes. Why did the authors consider the exchange terms in Section 2.4 and not the Kedem-Katalchsky’s law?

In our treatment, the assumption is that the diffusion constant of oxygen in the blood vessel wall is the same as that in the tissue. In the Kedem-Katalchsky’s law, precise experimental measurements of multiple parameters (e.g., thickness of blood vessel wall, vascular permeability, etc.) are used to evaluate the conductivity of oxygen through the blood vessel wall. This difference may not be significant for diffusion of nonpolar substances such as oxygen but may be important for some drugs. 

We can also easily incorporate the Kedem-Katalchsky law in our numerical method to evaluate the oxygen flux. We would be excited to test our model with more data from experiments in later applications. We have revised our manuscript and added relevant discussions in the manuscript (Line 122-133 and Line 180-181)

2.4.1 Section 3.2: Why do the authors use Finite Differences to discretize the diffusion reaction equation? It is a well-known fact that Finite Differences are locally not mass conservative, which is an important feature of a numerical method applied to a flow or transport problem.

The system in this paper contains no fluid convection and maintains a uniform diffusion coefficient. Under this situation, the central differential scheme is equivalent to the finite volume method, which ensures the local mass conservation. When convection becomes significant, we can use the mass-conservative finite volume method. The discussion are added on Line 213-217.

2.4.2 In Section 3.3 the authors suggest a post-processing method to reduce the errors caused by a coarse mesh. In this context, I would like to ask the authors to mention and briefly discuss a recent publication on the numerical modeling of 3D-1D coupled blood flow problems: T. Koch, M. Schneider, R. Helmig & P. Jenny (2020). Modeling tissue perfusion in terms of 1d-3d embedded mixed-dimension coupled problems with distributed sources. Journal of Computational Physics, 410, 109370.

These references are helpful for us. Instead of concentrated oxygen sources on the centerlines of blood vessels as in this work and many previous studies, distributed sources using smooth kernel functions are also used in the references. This distributed souce can be effectively used to avoid the weak singularities of the oxygen field. 

 We have added corresponding discussions on Line 156-158.

In this work, we aimed at obtaining relatively accurate solution with relatively large spatial step size, which prevent us from using distributed sources. Meanwhile, in our numerical method, since we also incorporate distributed sources, we also avoided possible consequences of the weak singularities.

2.5 Section 5.2: How do the authors compute the error between two levels? What norms are used? Furthermore I would suggest to report a suitable norm of the solution for several refinement levels. By this, the reader can observe on which mesh the numerical solution is representative. 

To avoid confusion, we have specify that the L2-norm for tissue oxygen partial pressure and oxygen sources are used in calculating the errors (caption of Figure 8). The errors based on these norms for different mesh sizes are shown in Figure 8 (a). 

2.6 Section 5.4, Figure 5: From my point of view the assignment of the labels a)-d) in the caption of Figure 5 are wrong. In a) you show the full network geometry, while in b)-d) the oxygen profiles for different retina depths are shown.

The reviewer is correct. We have revised accordingly. 

2.7 Section 6: How do the authors know that if the numerical error is below 1 %, it is smaller than the modeling error? How can the modeling error be quantified? Please provide a reference for this claim 

Although we believe that the modeling error is greater than 1% in general, we have not found suitable references for this claim. To avoid confusion, we have removed this statement. (Line 301, Line 455)

2.8 Appendix A: Lines 448/449: What are the boundary conditions and source terms for the Laplace equation? Do you consider the fundamental solution of the Laplace equation for the Taylor expansion?

We have added a detailed description for the evaluation of the 3D oxygen flux calculation, including the boundary conditions and source terms. This evaluation is based on series expansion solution of the Laplace equation under the cylindrical coordinate system. Due to the weak singularity of the solution for concentrated source on centerlines, this expansion is better than Taylor expansion. (Line 481-490)

2.9 Appendix B: Is the provided φ really continuous? Performing some computations reveals that the provided φ is not continuous at r = 1 and r = −1. Is the formula for φ correct?

We have corrected this typo in the manuscript. 

2.10 There are several typos and language errors. Thus I would recommend a proof reading to improve the quality of the written English.

We have reviewed our manuscript carefully revised the texts.

(See changes on Line 37, 94, 147, 173, 232, 344, 383, 387, 504, and in the caption of Figure 7)

---

## [Editor Report · Decision Letter 1]

11 Feb 2021

A fast numerical method for oxygen supply in tissue with complex blood vessel network

PONE-D-20-33706R1

Dear Dr. Dan Hu,

We’re pleased to inform you that your manuscript has been judged scientifically suitable for publication and will be formally accepted for publication once it meets all outstanding technical requirements.

Kind regards,

Adélia Sequeira, Ph.D

Academic Editor

PLOS ONE

Additional Editor Comments (optional):

The authors responded to all comments raised by the reviewers and, in my opinion, the paper can be accepted for publication in the revised form, with no further revision.
---

## [Editor Report · Acceptance letter]

15 Feb 2021

PONE-D-20-33706R1 

A fast numerical method for oxygen supply in tissue with complex blood vessel network  

Dear Dr. Hu:

I'm pleased to inform you that your manuscript has been deemed suitable for publication in PLOS ONE. Congratulations! Your manuscript is now with our production department. 

Kind regards, 

on behalf of

Dr. Adélia Sequeira 

Academic Editor

PLOS ONE